#### An algorithm for estimating the detection efficiency of a 1 lightning location system 2 Haibo Hu<sup>1</sup>, Xiya Zhang<sup>1</sup> 3 1. Institute of Urban Meteorology, CMA, Beijing 100089, China 4 5 Abstract This paper puts forward an algorithm for estimating the detection 6 efficiency (DE) of a lightning location system (LLS). The algorithm can be applied to 7 lightning flash/stroke density correction (e.g., cloud-to-ground (CG) lightning 8 flash/stroke density) and LLS performance evaluation. A lightning strike density 9 correction for DE promotes the applicability of the LLS data. Fundamentally, the 10 generalized extreme value (GEV) distribution was found to best fit the probability 11 distribution of the signal strengths of the lightning observed by the ADTD detectors in 12 13 Beijing, China. With respect to this probability distribution, we estimated the single-station acceptance damped by the uneven underlying land surface conductivity. 14 15 Accounting for the multi-detector location modes supported by single-station 16 acceptance, the iterative algorithm was applied for deducing the DE of a LLS. In this case study, the DE estimates of the ADTD network were lower in the mountainous 17 18 areas than in the plains. These lower estimates can be due to the low underlying 19 conductivity of the mountainous areas, which creates a high attenuation effect on the lightning electromagnetic signals, and the greater distances from the lightning 20 detectors. Subsequently, the cloud-to-ground (CG) lightning flash/stroke density 21 22 derived from the ADTD data was corrected for the DEs. The results indicated that the CG lightning flash/stroke densities in the northern and northwestern mountainous 23 areas are lower than that in the highly urbanized plains. This anomaly is due to the 24 effects of the increased roughness of the underlying land surfaces, enhanced aerosols, 25 urban heat island (UHI), and intensifying thunderstorm activities in urban areas, but 26 this anomaly is not likely related to the DE discrepancy. 27 28

- 1 -

29 Key words Lighting location system, Detection efficiency, Cloud-to-ground lightning

- 2 -

1 flash/stroke density.

## 2 Introduction

3 Lightning location systems (LLSs) capture natural lightning electromagnetic 4 signals and consequently locate lightning sources. LLS observations can be used to derive and retrieve lightning parameters that indicate lighting characteristics, e.g., the 5 cloud-to-ground (CG) lightning flash density, the positive/negative CG flash/stroke 6 7 density, the percent of positive/negative flashes, the first stroke negative and positive peak currents, and the multiplicity for negative and positive flashes (Orville et al. 8 2002; Orville et al. 2001; Rudlosky and Fuelberg 2010; Taszarek and Czernecki 2015). 9 These types of data have been widely used in weather forecasting, severe weather 10 warning systems and climatological analysis, e.g., assimilation in weather forecasting 11 models (e.g., Fierro et al. 2011; Fierro and Reisner 2011), synoptic analysis of 12 thunderstorms (e.g., Changnon 1993; Cummins and Murphy 2009), presentations of 13 14 lightning characteristics (e.g., Schulz et al. 2005; Drüe et al. 2007; Mäkelä et al. 2010), lightning climatology and climate pattern recognition (e.g., Antonescu and Burcea 15 2010; Williams et al. 2005; Taszarek et al. 2015; Etherington and Perry 2017), and 16 17 lightning risk assessment (Hu et al. 2014). However, a serious problem emerges when a LLS provides an uneven detection efficiency (DE) within the detectable distance or 18 19 even a very low DE at the edges of the detectable distance (Naccarato and Pinto 2009), 20 which greatly restricts the applicability of the LLS data (Hu et al. 2014; Yao et al. 2012). 21

DE is determined by the performance and sensitivity of the sensors, the sensor network geometry, the underlying ground conductivity (Schütte et al. 1988; Naccarato and Pinto 2009; Mäkelä et al. 2010), etc. The unevenly distributed DEs contribute to LLS data uncertainty and ruin data validity (Williams 2005; Yao et al. 2012; Hu et al. 2014). Thus, it is desirable to further develop algorithms to estimate the DE, which can be applied for correcting the lightning flash/stroke density derived from the LLS data.

Moreover, such an algorithm is useful in LLS performance evaluation and network deployment validation and would help to determine optimal detector

Natural Hazards and Earth System Sciences Discussions

- 3 -

1 deployment (Cummins et al. 1998; Idone et al. 1998). This methodology was originally intended to test network deployment optimization and network performance 2 and eventually became applicable to improve LLS data quality. Schütte et al. (1987, 3 4 1988) suggested that the lightning impulse signal strength (LISS) weakens linearly with propagation distance and confirmed the Weibull probability distribution of LISSs 5 received by a single station within a detectable distance. Naccarato and Pinto (2009) 6 7 estimated the DE of the Brazilian lightning detection network (BrasilDAT) based on the individual sensor DE probability functions, which were derived from a large 8 amount of CG stroke data provided by the network. 9

These applicable approaches obviously possess reference values in the aftermath 10 of DE estimation. Note that the precedent work can be cited as a referential source; 11 however, it is inappropriate to simply copy the methods of earlier researchers if 12 considering different data resources and circumstances. For example, the Weibull 13 14 distribution of LISSs identified by Schütte et al. (1987, 1988) only reflected the lightning characteristics observed by the European LLS network at that time. 15 However, our study revealed that the probability distribution of the LISSs detected by 16 17 the ADTD sensors deployed around Beijing, China, would fit the generalized extreme value (GEV) distribution, rather than the Weibull distribution. This finding 18 19 demonstrates the necessity of modifying the precedent methodology to estimate 20 single-station acceptance.

Moreover, the DE and location accuracy of a LLS can be influenced by the 21 lightning location mode, which is the number of detectors synchronously receiving 22 23 the lightning electromagnetic signal in one lightning location. The algorithm should consider the multi-detector location mode related to the number of detectors in the 24 lightning location. Usually, at least two magnetic direction finder (MDF) detectors 25 and three time-of-arrival (TOA) detectors are needed in each lightning location 26 27 (Schütte et al. 1988). As observed by a LLS network with newly upgraded IMPACT detectors (combined MDF and TOA technology), a lightning source can be located 28 with signals synchronously detected by two or more such detectors (Bourscheidt et al. 29 2012). With multi-detector location modes, here, an iterative algorithm is introduced 30

- 4 -

1 to derive the DE from single-station acceptance.

Underlying surface conductivity plays an important role in damping lightning signals and eventually influences the DE (Honma et al. 1998; Cummins et al. 2005; Rudlosky and Fuelberg 2010). Schütte et al. (1987) suggested that land surface damping influences both the DE and location accuracy and that the TOA network appears more sensitive to land surface damping. At this point, the algorithm should consider the damping effect on signal strength, which is induced by land surface conductivity.

The algorithm was constructed by following the abovementioned fundamentals. 9 In this methodology, the LISSs were calculated using a linear signal propagation 10 model suggested by Cummins et al. (1995). The damping coefficients in a grid system 11 were estimated by addressing the land surface conductivity of specific land use and 12 land cover (LULC) types. After correcting the signal strength magnitudes of the 13 14 damping factor, we identified the GEV probability distribution of the LISSs. Based on this probability density function (PDF), we estimated the single-station acceptance of 15 lightning source signals and ultimately calculated the grid DEs using an iterative 16 17 algorithm. We corrected the CG lightning stroke density for the DEs and found an improvement in the estimates of the CG lightning stroke density in comparison with 18 19 the uncorrected density.

Admittedly, the uncertainty cannot be reduced completely through DE estimation because the DE estimation fundamentals involve simulating signal attenuation propagated over certain distances and dampening due to land surface conductivity. The improvement in the LLS data quality (e.g., DE and location accuracy) basically depends on the implementation of optimal detector deployments and detector upgrading, and DE estimates are critical in promoting the applicability of the LLS data.

#### 27 **1. Data description**

The LLS data collected from 2007-2016 by the ADTD (Advanced TOA and Direction system, where TOA denotes time-of-arrival) network deployed by the China Meteorology Administration (CMA) were used to 1) identify the probability

C Author(s) 2

- 5 -

distribution of the LISSs and 2) derive the CG flash/stroke density, which is corrected
 for the DE. These data include the time, location, amperage and polarity of the CG
 lightning strokes.

4 The ADTD consists of more than 301 sensors (as of March 2011) in China (Yao et al. 2012). In Beijing, 9-14 ADTD-1 sensors (improved IMPACT (combined MDF 5 and TOA) sensors) can detect 1-450 kHz (the low-frequency (LF) band) lightning 6 7 sources (Fig. 1). The ADTD-1 sensors use the combined MDF and time-of-arrival (TOA) method for position retrieval (Ma 2015). In this method, if a lightning source 8 is only detected by two ADTD-1 sensors, the algorithm uses one TOA hyperbolic 9 curve and two MDF vectors to retrieve the position. If a lightning source is detected 10 by three sensors in a non-duplicate region, the TOA algorithm is used to retrieve the 11 position directly. In contrast, if the TOA is the first to find a duplicate location, then 12 the MDF method is used to identify the true location. If a lightning event is detected 13 14 by four or more sensors, a TOA least squares method is used to find a more precise location. Thus, a lightning location detected by four or more sensors is more precise 15 than that detected by fewer sensors. In the ADTD data, the percentage of lightning 16 17 sources reported by four or more sensors relative to the total number of detected sources is 63.3%. Meanwhile, the ADTD-observed +CG and -CG lightning peak 18 19 currents are in the ranges of 0.08 kA to 995.9 kA and 0.258 kA to 992.6 kA, 20 respectively (Fig. 1).

The ADTD sensor manufacturers claim that the DE of the ADTD sensors could 21 be 90% at distances between 300 and 600 km, with a median location accuracy error 22 23 of 1 km. However, only the flash DE can be 90%, and the stroke detection efficiency (SDE) is lower. The first stroke peak current in a multiple-stroke CG flash can be 24 greater than twice its subsequent stroke peak current (Rakov and Uman 1990). Thus, 25 the sensors can capture the first larger peak stroke but miss the weaker subsequent 26 27 stroke (Rudlosky and Fuelberg 2010). Moreover, some weak CG strokes (including single-stroke CG flashes) cannot be detected due to the signal attenuation induced by 28 long-distance propagation and damping effects in lower conductivity mountainous 29 regions (Schütte et al. 1988). 30

- 6 -

We estimated the SDEs of the ADTD in a grid system (1 km × 1 km, see Fig. 2) and corrected the lightning stroke density using the SDE. The SDE estimates approximate those of the U.S. NLDN (National Lightning Detection Network) in 1998, which was reported to be 62% (Idone et al. 1998). Hence, the DE level of the ADTD is equivalent to that of the NLDN, at least in 1998, suggesting that a considerable potential for improvement remains in terms of network upgrades and that DE estimation is necessary for promoting the applicability of the ADTD data.

11 Fig. 1 Histogram of the peak current probability density of (a) the –CG lightning, (b)

- 7 -

- 1 +CG lightning and +IC lightning identified from their peak currents less than 15 kA
- 2 and (c) the total CG lightning.

### 3 **2. Method**

The probability distribution of the LISSs is critical in estimating single-station acceptance. Being prior probability distribution, it can be deduced using samples of lightning location data. The likely probability distribution of these data would be that of either the Weibull, generalized extreme value (GEV), lognormal (LN), extreme value (EV) or gamma distribution. Hereinafter, we primarily introduce the GEV distribution since it would provide the most accurate probability distribution fit of the LISSs in the Beijing region.

#### 11 2.1 GEV distribution

12 The GEV distribution has a cumulative distribution function (CDF)

13 
$$F(x \mid \mu, \sigma, \xi) = \exp\{-[1 + \xi(\frac{x - \mu}{\sigma})^{-1/\xi}]\}$$
 (1)

for 1 + ξ(x - μ) / σ > 0, where μ ∈ R is the location parameter, σ > 0 is the
scale parameter and ξ ∈ R is the shape parameter. Thus, for ξ > 0, the CDF is
only valid when x > μ - σ / ξ; for ξ < 0, the CDF is valid and has a short-tailed</li>
Gumbel distribution when x < μ + σ / (-ξ). For ξ = 0, the CDF is formally</li>
undefined and can be replaced by the result obtained by taking the limit as ξ → 0.
Then,

20 
$$F(x \mid \mu, \sigma, 0) = \exp\left[-\exp\left(-\frac{x-\mu}{\sigma}\right)\right]$$
(2)

21 with no restriction on x.

22 The density distribution function is consequently

23 
$$f(x \mid \mu, \sigma, \xi) = \frac{1}{\sigma} [1 + \xi(\frac{x - \mu}{\sigma})^{(-1/\xi) - 1} \exp\{-[1 + \xi(\frac{x - \mu}{\sigma})]^{-1/\xi}\}$$
(3)

- 8 -

- When  $\xi = 0$ , the density is positive over the entire real line and equal to the Weibull 1
- 2 distribution:

3 
$$f(x \mid \mu, \sigma, 0) = \frac{1}{\sigma} \exp\left[-\left(\frac{x-\mu}{\sigma}\right)\right] \exp\left\{-\exp\left[\left(-\frac{x-\mu}{\sigma}\right)\right]\right\}$$
(4)

4

#### 2.2 Fundamentals of the GEV distribution applied for estimating single-station 5 acceptance 6

Consider a normalized (for example, to a distance of 100 km), undamped 7 reference signal-strength distribution with the frequency function  $f(s_0)$  and an 8 expression for the signal strength dependence on the distance  $s_0 = D^{-1}(s)$ . The simplest 9 case for an undamped radiation field is the inverse power law  $s=s_0r_0/r$ , where  $s_0=sr/r_0$ . 10 The acceptance can be calculated as a function of the distance and the lower and 11 12 higher thresholds  $s_{min}$  and  $s_{max}$ , respectively:

13 
$$A(r) = \int_{sD_{\min}}^{sD_{\max}} f(s_0) ds_0, s_0 = D^{-1}(s), ds_0 = D^{-1'}(s) ds$$

14 
$$= \int_{sD_{\min}}^{sD_{\max}} f[D^{-1}(s)]D^{-1'}ds = F[D^{-1}(s)] \Big|_{S_{\min}}^{S_{\max}}$$
(5)

15 For undamped propagation, we can obtain

16 
$$A(r) = F(sr/r_0) \Big|_{s_{\min}}^{s_{\max}}$$
(6)

In the case of the GEV distribution, the acceptance is given by 17

$$A(r) = \begin{cases} 0 & r < cr_0 / s_{max} \\ \exp\left\{\left[-\left[1 + \xi\left(\frac{S_{max}r / r_0 - \mu}{\sigma}\right)^{-1/\xi}\right]\right\} & cr_0 / s_{max} < r \le cr_0 / s_{min} \\ \exp\left\{\left[-\left[1 + \xi\left(\frac{S_{min}r / r_0 - \mu}{\sigma}\right)^{-1/\xi}\right]\right\} - \exp\left\{\left[-\left[1 + \xi\left(\frac{S_{max}r / r_0 - \mu}{\sigma}\right)^{-1/\xi}\right]\right\} & r > cr_0 / s_{min} \end{cases} \end{cases}$$

$$19 \qquad (7)$$

where c is the signal bias and can be replaced by the minimum value of the signal 20 21 samples.

22 The effective radius,  $\rho$ , describing the properties of a lightning counter or

1 direction finder can be defined as  $\pi \rho^2 = 2\pi \int_{-\infty}^{\infty} A(r) r dr$ 2 (8) 3 This definition means that an ideal lightning counter accepts all lightning s with  $r \le \rho$  and none with  $r > \rho$ . 4 5 When the signal strength is GEV distributed and the counter has as a sufficient dyn