# Peer review of "An algorithm for estimating the detection efficiency of a 1 lightning location system 2 Haibo Hu1, Xiya Zhang1 3 1. Institute of Urban Meteorology, CMA, Beijing 100089, China 4 5 Abstract This paper puts forward an algorithm for estimating the detection 6 efficiency (DE) o"

_Natural Hazards and Earth System Sciences, 2017_

## Short Comment (SC1) · 27 Nov 2017

General Comments This manuscript describes an algorithm and method for estimating the detection efficiency of a lightning location system for cloud-to-ground return strokes, and applies this method to assess the performance of the Advanced TOA and Direction System (ADTD) LLS in the vicinity of Beijing, China. The authors present their material in a logical way, with generally good organization. However, this reviewer has concerns about several aspects of the approach, and about the value of constraining the lightning signal strength distribution using a parametric probability distribution. The comments below reflect the key concerns that call into question the methods and results presented in this work. Other lesser but important comments will be provided if the authors are able to address these first issues.

[Figure]

To this reviewer, this work is primarily a re-statement of the earlier work by Schütte, using a more-general parametric probability distribution. It suffers from the same limitations as that earlier work, as expressed in the first two numbered comments in this review. I do not see new and/or novel concepts, ideas, tools, or methods. I would be happy to be convinced by the authors to change my opinion.

Specific Comments

1. My most fundamental concern about this work is what appears to be the use of a single "acceptance" function A(r) that is independent of the peak current of the underlying lightning discharge. This is not a problem for single-sensors flash counters, but it is a problem when evaluating detection that requires simultaneous measurements from more than one sensor. Consider a 2-sensor network where the sensor thresholds are the same. Then consider a lightning location that is close to one sensor and far from the other one. The close sensor will not report high-current discharges because their signal exceeds Smax (saturation), and the distant sensor will not report low-current discharges that fall below the minimum detection level (Smin). Under such a condition, one sensor would report only low current discharges not reported by the other sensor, but the A(r) values associated with that location, as defined in this work, will be non-zero for both sensors. This will result in a non-zero detection efficiency value for the two-sensors network. In the worst case, each sensor could have a DE of 0.50 at some distance r, with no overlap in the reported peak current. In this case, the model-computed DE for that location would be 0.25, but the actual DE would be zero. Some modern DE models avoid this problem by defining their equivalent to the A(r) values for narrow ranges of peak currents, and them summing their contributions after weighting by the probability of occurrence of discharges in each narrow range or currents (such as Naccarato and Pinto, cited in this work). Another approach is to partition the DE Model into three elements (peak current distribution, radio propagation/losses, and sensor threshold functions) and then computing DE from these physical characteristics, as described in an appendix in Pessi et al., 2009 and refined in an appendix to

[Figure]

Nag. et al., 2015).

2. In the first key point in the Conclusions of this work, the authors state that "It is critical to identify a suitable probability distribution if the LISSs", and they go on to discuss various parametric distributions. I question both the need for, and value of, employing a parametric distribution (in this case, the generalized extreme value (GEV) function) to describe the distribution of "lightning impulse signal strengths" (LISS) values. Since the underlying LISS distribution is derived through measurement using a LLS, presumably in a region with very high DE, the necessary cumulative distribution is simply the integral of the measured distribution. I do see that the authors have employed the GEV formulation to produce formulas for A(r) and an effective radius, but neither of these are required to compute detection efficiency. I ask that the authors acknowledge that a parametric distribution is not ESSENTIAL for DE calculations, and then state the value that they fell comes from using one.

3. Equation 7 seems to have a problem, at least in terms of the range of valid values for the three expressions. It might just be that the meaning of the variable "c" is not correct, or that I simply do not understand it. The text refers to "c" as the "signal bias" (which needs to be defined), but then it says that it can be replaced by the "minimum values of the signal samples" (which is confusing, since there are two "signals" – the LISS signal, and the propagated signal received by the sensors). I would expect A(r) to have distinct behavior over 5 separate ranges of range r, as depicted in Schütte (JAOT, September 1987), Figure 1:

a. Small r ( < 10 km), where signals from even the lowest peak current (LISS) will saturate the sensors so A(r) = 0;

b. Somewhat larger values of r (∼10-50 km), where signals from the lower currents do not saturate the sensor, but where the large ones do;

c. Intermediate values of r (50-200 km for lamda = 0.3) where the signals from all currents are above Smin and below Smax. In this case, the acceptance is 1.0 (100%)

d. Large values of r (200-500 km for lamda = 0.3) were signals from progressively larger currents fall below the lower (Smin) threshold as the range r gets larger; and

e. Very large values of r (>500 km for lamda > 0.3) where signals from all currents fall below the lower threshold, and A(r) = 0.

It might be very helpful for the reader if the paper included a representative plot of A(r).

4. The authors' handling of Damping in section 2.4 and 3.2 is not very clear, and it may have technical errors. I am left with the following questions and problems:

a. What is the source reference (paper) for the damping function and parameters, and why were they selected?

b. Section 3.2 implies that more than 1 conductivity value was used, but equation 15 indicates a single value

c. If different conductivity values were used in different regions, how was this applied to the propagation equation? Is there a theoretical basis for how this was done?

d. There seems to be two different meanings for "sigma" in Equation 17 – one of for the ground conductivity, and one is a GEV parameter. This is a real problem.

e. In Section 3.2, the authors refer to values with units of Ohm-meters as conductivity, but they are actually resistivity.

5. Equation 19 for F(3,2) does not seem to be correct. The first term seems to be associated with all three sensors "accepting" the event. Also, the definition of A(r2) in the text that follows this equation does not make sensor – how is it the acceptance of detector r3 given the value of A(r2) when A(r2) is already used on equation 18 for a 2-sensor network?

NOTE: I will attempt to send the three new references mentioned in this review. Thus far, I was only able to attach one of the three references to this online review

Please also note the supplement to this comment:
https://www.nat-hazards-earth-syst-sci-discuss.net/nhess-2017-307/nhess-2017-307-SC1-supplement.pdf
* * *
[Figure]

**Supplement:**

[supplement omitted: unrelated document]

---

## Referee Comment (RC1) · Anonymous Referee #1 · 28 Nov 2017

The paper discusses detection efficiency of a lightning detection network in the Bejing area. It presents many interesting observations so that a publication can be envisaged. However, there are many shortcomings that must be remedied before publication. First, many procedures and evaluations are not described appropriately and remain diffuse. Second, the authors ignore the quality of now-a-days networks and remain oriented along very old networks. Thus, substantial revision is necessary.

In the following a list of obvious detail-problems is given.

p. 1, line 7 please define the detection efficiency DE; there are several possibilities to do so.

p. 1, line 14 the term "single-station acceptance" is not defined. Please clarify.

[Figure]

p. 1, line 15/16 it is not generally known what the authors understand by multi-detector location modes

p. 1, line 17/27 the term "lower" does not mean much; please quantify. Also, it is normal that DE varies in a network, especially versus and beyond the border lines; it should be said that the intrinsic network DE as regards sensors and baselines is not responsible for the observed variations of DE – if this is true.

p. 2, line 22 it is unclear what the authors mean by "performance". Please specify.

p. 2, line 23 it is known that ground conductivity affects signal attenuation, but this is not a very dramatic effect, in any case not comparable with sensor and network features dominated e.g. by noise and thresholds. The authors should give the km-extension of the considered network area. It appears that all conclusions come from within some 600 km, where other networks operating elsewhere did not find large effects of the claimed type.

p. 3, line 10-20, 13ff this paragraph is not understandable. It is not clear what network or lightning parameters enter into the quoted statistical distribution and how this relates to the (undefined) single-station acceptance.

p. 3, line 21-30 this paragraph can be deleted or should be drastically updated because the given information is outdated and does not account for modern high-precision networks. In particular, it does not take into account the large difference of DF and TOA in terms of location accuracy (LA).

p. 4, line 5 the wording "TOA network appears more sensitive to land surface damping" is misleading. Signal damping has nothing to do with the kind of signal evaluation at a later time (DF, TOA, or both). There are many highly accurate networks that do not take into account damping and still obtain quite homogeneous DE and high LA.

p. 4, line 18 please specify the improvement

p. 4, line 24/25 please specify "optimal" and "upgrading". What are the deficiencies

and what measures are taken to reach what kind of improvement?

p. 5, line 11 please specify "non-duplicate region"

p. 5, line 19/20 it seems quite strange to quote "0.08 kA to 995.9 kA and 0.258 kA to 992.6 kA" ! First, stroke currents as low as given hardly exist, and second, the high numbers are outside any verified range, and suggest a totally meaningless accuracy.

p. 5, line 22 what is the meaning of 90% DE? How is the definition for 100% ? In view of the reported extremely small CG currents of 0.08 kA (a CG stroke that small was never ever reported anywhere else) it can be virtually excluded that the quoted %-DE values have high credibility. There seems to be a big mix of definitions, observations, and interpretations. The entire data descriptions needs substantial revision.

p. 5, line 23 what is the reason for the statement "only the flash DE can be 90%" ? This statement is not understandable.

p. 5, line 24 the authors still follow the outdated claim that the first stroke of a flash is the most intensive one. This is not correct, because statistics show that flashes with high multiplicity exhibit the strongest stroke in the 2nd – 4th stroke order.

p. 6, Fig. 1 please define the scale "probability density". Fig. 1 exhibits the most likely current around 24 kA. This is best proof that the network DE is extremely low. Modern high-quality networks have the maximum far below 10 kA, mostly between 4 and 6 kA. For this reason, all the %-DE values in the present paper are very suspicious.

p. 8, line 12 it may be best to give the explanation for the definition of "single-station acceptance" in this chapter.

p. 11, line 4 how is acceptance confirmed? Why is an iteration needed?

p. 12, line 6/7 in 10 years 240,804 CG strokes are reported. What was the area? Is the resulting flash density reasonable?

p. 12, line 10 what is a "certain" threshold? Please quantify.

p. 12, Fig. 2 what is plotted? The x-axis (hardly to identify) seems to scale the signal strength (in a.u.?); the y-axis gives a "probability density"; please specify the unit: strokes/a.u. or strokes/kA or what else?

p. 13, line 7-12 other networks are not comparable and not relevant here, please delete

p. 13, line 24 please explain "LCLU"

Fig. 3-5: the km-scale must be larger, it is hard to read in the graph; may be shifted to the caption (?)

p. 16, line 13 as stated above a DE of 78% is not possible when the most likely stroke current is as high as 24 kA

p. 16, line 14-20 comparisons with other networks are not relevant, please delete

Table 2 The trend of the numbers of used sensors is highly suspicious and unreasonable. Why should it be easier to locate with 6 rather than 5 sensors? Please explain the remarkable discontinuity in the number of used sensors (e.g. 37.6% for 6 sensors)

p. 21, line 2 is there any quantitative result about the minimum threshold Smin ?

---

## Referee Comment (RC3) · K. Cummins (Referee) · 4 Jan 2018

General Comments

(provided in initial review)

Less-critical comments:

1. The term "single-station acceptance" is used in the abstract and the Introduction, but the reader has to go through the mathematics of the model (equations (5) and (6)) to begin to understand what it means. It seems that this term is derived from the work of Schutte. There are assumed-perfect thresholds (smin and smax) that are implicit in this model. The authors should provide a brief overview of the physical processes that impact the detectability of lightning, and then define "single-station acceptance" in that

context.

2. Pg 2; line 13: The Cummins and Murphy reference does not seem to be appropriate in the context of synoptic analysis of thunderstorms.

3. Pg 2; line 25: "ruin" seem to be too strong of a word. I suggest "compromise."

4. The paragraph starting on line 9 of page 4 partially describes the authors' model. This does not seem to belong in the Introduction section. Maybe there should be a "Modeling" section starting here, and it could also include the description requested in the previous comment.

5. Pg 4; line 16: The authors define their DE calculation as being "iterative." Typically, this term is only used describe methods that do not have a known and finite number of steps. In the case of DE modeling when there are a know number of combinations of sensors and "location modes" the term "combinatorial" is more accurate.

6. Pg 5; line 6-7: the authors are talking about "lightning sources", but then they reference figure 1 which is a set of "peak current probability distributions". They should explain the relationship between "lightning sources" (an ambiguous term) and "peak current." It would also be good to tell the reader that the peak current distributions are really LISS distributions, scaled in a manner to that converts the range-normalized magnetic (or electric) field to an estimate of the current near the base of a lightning channel during a CG return stroke. In addition, the authors mention a 15 kA value in the caption to Figure 1, with no explanation. Many readers will not know the reason for this limit, so it should be explained, or a reference should be provided in the text.

7. Pg 5; lines22-23: an individual sensor does not have a specific location accuracy. The location accuracy of the network at a point-of-interest is determined by the timing- and angle-errors of the individual sensors, and the geometry of the sensors, relative to that point-of-interest.

8. Pg 6; first line: This paragraph seems to belong in a discussion or conclusions

section.

9. Bottom of pg 8: the authors define effective radius as if it is their definition. I think that it was defined in 1987 by Schutte (possibly earlier), and used in his development of the "Acceptance" concept. This should be acknowledged.

10. Bottom of Pg 9: The authors say that equation 13 describes a linear relationship between Ip and s, then they show (in equation 13) a non-linear dependence on r. It might be helpful to clarify that this is true after s is corrected for propagation "s".

11. Pg. 10 line 16: use of the variable "D" (for damping) could confuse some readers, since D is first used in equation 5. The D in equation 5 itself is somewhat confusing, since it implies a function mapping through its inverse.

12. Figure 2: The probability distribution shape-matching illustrated in Figure 2 is of concern, since there is a sampling problem – note that the curves try to match the large value "envelope" of the histogram, but those values are surrounded by much lower (sometime zero) values in adjacent histogram bins.

13. Figure 3 and 4: what is the meaning of the dark-red colored region in the lower-right portion of figure 3(c) and 4? They are not colors in the color table.

14. Pg. 16, line 4: The Acceptance does NOT clearly decrease towards the mountains – it seems to decreases in all areas at similar distances from the sensors due to increasing range.

15. Pg 19, lines 13-15: This conclusion does not seem to be justified from the content in this paper.

16. Figure 5: the corrected stroke density neat Guangxiangtai seems too low, given the significant drop-off if DE near that site shown in Figure 4(b). Which DE correction was used to produce this figure? End

2017-307, 2017.
Interactive
comment